# LISTEN TO MOTION: ROBUSTLY LEARNING CORRELATED AUDIO-VISUAL REPRESENTATIONS

## ABSTRACT

Audio-visual correlation learning has many applications and is pivotal in broader multimodal understanding and generation. Recently, many existing methods try to learn audio-visual contrastive representations from web-scale videos and show impressive performance. However, these methods mainly focus on learning the correlation between audio and static visual information (such as objects and background) while ignoring the crucial role of motion information in determining sounds in videos. Besides, the widespread presence of false and multiple positive audio-visual pairs in web-scale unlabeled videos also limits the performance of audio-visual representations. In this paper, we propose **Li**sten to **Mo**tion (LiMo) to capture motion information explicitly and align motion and audio robustly. Specifically, for modeling the motion in video, we extract the temporal visual semantic by facilitating the interaction between frames, while retaining static visual-audio correlation knowledge acquired in previous models. To prompt a more robust audio-visual alignment, we propose learning motion-audio alignment more specifically by distinguishing different clips within the same video. And we quantitatively measure the likelihood of each sample being false positive or containing multiple positive instances, then adaptively reweight samples in the final learning objective. Our extensive experiments demonstrate the effectiveness of LiMo on various audio-visual downstream tasks. On audio-visual retrieval, LiMo achieves absolute improvements of at least 15% top1 accuracy on AudioSet and VGGSound. On our newly proposed motion-specific tasks, LiMo exhibits much better performance. Moreover, LiMo also achieves advanced accuracy on audio event recognition, demonstrating enhanced discriminability of audio representations.

## 1 INTRODUCTION

Audio and visual modalities are naturally correlated in the real world. The audio-visual correlation learning plays an important role in multimodal understanding (Zhao et al., 2023; Zhang et al., 2023; Su et al., 2023; Chen et al., 2021b; Senocak et al., 2018; Hu et al., 2019; Huang et al., 2023) and generation (Lee et al., 2023; Ruan et al., 2023; Tang et al., 2023), and has a wide range of applications, such as sound effect matching or generation, discordant audio detection, and sounding video generation.

Recently, many methods (Arandjelovic & Zisserman, 2018; Rouditchenko et al., 2020; Gong et al., 2022; Girdhar et al., 2023; Wang et al., 2023a) try to capture the audio-visual correlation by learning high-quality audio-visual contrastive representation from web-scale unlabeled videos (Gemmeke et al., 2017; Miech et al., 2019; Chen et al., 2020). Despite their impressive performance, two key issues still limit the further development of audio-visual representations: 1) Previous methods mainly model the static "object" information from a few frames while lacking the ability to capture and align the important temporal "motion" information. However, both visual "object" and "motion" information play pivotal roles in learning audio-visual correlation. The former indicates videos of different objects may sound different, while the latter means different actions of the same object also result in different sounds. 2) The unlabeled web-scale video data are noisy. In a video, the visual information is limited in the camera perspective, while the audio can originate from all directions. Consequently, not all visual objects make sounds, and not all sound sources are visible in the video. This unavoidable noisy data compromises the quality of the learned representations.

This paper proposes **Li**sten to **Mo**tion (LiMo), a novel audio-visual representation learning framework to address the above limitations. We enable the visual encoder to capture the temporal "motion" information, while retaining the learned correlation between the static "object" information and audio in the pre-trained model. Besides, we further introduce a motion-audio alignment with samples reweighting to deeply and robustly learn the correlated audio-visual representations. To align motion and audio more specifically, we employ a clip-level contrastive loss, which considers both clips from different videos and different clips from the same video as negatives. Since the primary visual differences among clips from the same video lie in their temporal motion, clip-level contrastive loss prompts a deeper understanding of the correspondence between motion and audio. To alleviate the influence of noisy data and achieve a robust learning process, we propose to quantitatively measure the likelihood of each sample being a false positive and containing multiple positive instances. To this end, we calculate the audio-visual matching confidence of videos and the distinguishability scores of clips from the same video. By reweighting the samples in the final contrastive loss calculation process, we can effectively diminish the detrimental effects of noisy data.

Our experiments demonstrate the effectiveness of LiMo on various audio-visual tasks, including general audio-visual retrieval, newly proposed motion-specific audio-visual tasks, and audio event recognition. On audio-video retrieval, LiMo achieves absolute improvements of at least 15% top-1 accuracy on AudioSet (Gemmeke et al., 2017) and VGGSound (Chen et al., 2020) compared to other advanced audio-visual models. In addition, LiMo shows significant advantages over previous methods in the newly proposed audio-based video grounding and lip-speech retrieval, which focus on the correlation between motion and audio. For event classification tasks, the robust motion-audio alignment also enhances the discriminability of audio representations.

Our contributions can be summarized as three-fold: 1) We propose **Li**sten to **Mo**tion (LiMo), a new audio-visual pre-trained framework that emphasizes the importance of visual motion information in audio-visual learning. 2) We propose an adaptive reweighted contrastive loss, which effectively mitigates the adverse effect of the ubiquitous noisy data in web-scale unlabeled video data. 3) We conduct extensive experiments on multiple audio-visual downstream tasks and datasets to showcase LiMo's state-of-the-art performance and validate our design's effectiveness. Moreover, we further propose two motion-specific audio-visual tasks to more specifically verify the correlation between visual motion and audio.

## 2 RELATED WORK

### 2.1 AUDIO-VISUAL REPRESENTATION LEARNING

Audio-visual representation learning aims to pre-train models on large-scale visual-audio pairs extracted from web-scale unlabeled videos. The learned representation in such a pre-trained model can capture the correlated semantics of audio and visual modalities. Inspired by the success of vision-language representations (Radford et al., 2021; Jia et al., 2021; Li et al., 2021), most recent works follow contrastive learning schemes to pre-train audio-visual models. AudioCLIP (Guzhov et al., 2022) and WAV2CLIP (Wu et al., 2022) leverage audio-image-text pairs from AudioSet (Gemmeke et al., 2017) to train an extra audio encoder for vision-language pre-trained model, while C-MCR (Wang et al., 2023b) establishes an audio-visual representation space by connecting the pre-trained CLIP (Radford et al., 2021) and CLAP (Wu et al., 2023) space through text. AVST (Chen et al., 2021a) tries to learn contrastive audio-visual alignment from videos of the VGG-Sound (Chen et al., 2020) dataset. CAV-MAE (Gong et al., 2022) combines the contrastive learning with masking data modeling (Devlin et al., 2018; He et al., 2022; Huang et al., 2022) and further improves the performance on audio-visual downstream tasks. ImageBind (Girdhar et al., 2023) collects data of multiple modalities (including audio) paired with images and binds these different modalities to CLIP space via contrastive loss.

Although these methods achieve promising performance on different audio-visual downstream tasks, they either lack modeling of visual motion information or explicit learning of motion audio alignment, which significantly constrains their upper bound in capturing audio-visual correlations. Besides, the visual information in the video is solely derived from the camera perspective, whereas the audio can originate from any source. Thus, the web-scale video data for training is very noisy. However, previous audio-visual representation methods lack the analysis and design to alleviate the adverse effect of the noisy data.

## 2.2 CONTRASTIVE LEARNING FROM NOISY DATA

Multimodal contrastive pre-training requires millions or even billions of level data pairs collected automatically from the Internet (Bain et al., 2021; Changpinyo et al., 2021; Schuhmann et al., 2022). Despite certain trivial pre-processing methods for data cleaning, the noise in these unlabeled datasets remains the major problem in multimodal contrastive learning. In the vision-language field, image-language pre-training also suffers from noise in datasets, and some works attempt to mitigate the negative impact of noisy samples. ALBEF (Li et al., 2021) maintains a momentum model and utilizes the more stable predictions of the momentum model as additional supervision. On the other hand, BLIP (Li et al., 2022) bootstrapping generates novel captions for images and filters out noisy samples. LiT (Zhai et al., 2022) keeps the visual encoder frozen to preserve well-learned visual representations from being affected by imperfect language supervision. Compared to image-text data, audio-visual data (Chen et al., 2020; Gemmeke et al., 2017) is even more noisy, as its weak correlations are more common and harder to detect. Besides, videos often contain hard-to-distinguish motions (tiny movements or even static video) and audio (repetitive rhythms or even silence). These samples cannot provide meaningful motion-audio correlation information and will affect the stability of motion-audio alignment learning.

## 3 METHOD

In this section, we first revisit contrastive learning and analyze why it is susceptible to noisy data. Then, we introduce the inputs and architecture of LiMo. Lastly, we describe the robust Clip-level audio-visual alignment loss with adaptive samples reweighting method.

### 3.1 REVISITING CONTRASTIVE LEARNING

Contrastive learning showcases impressive achievements in multimodal representation learning. Considering $N$ paired data from two different modalities, each pair is encoded to $\mathbf{x}_i, \mathbf{z}_i$. Contrastive learning pulls paired data's features closer, pushing unpaired data away, leading to a discriminative multimodal representation space. The general multimodal contrastive learning loss can be formulated as:

$$L = -\frac{1}{2}\frac{1}{N}\sum_{i=1}^{N}\left[\log\frac{\exp(\text{sim}(\mathbf{x}_i, \mathbf{z}_i)/\tau)}{\sum_{j=1}^{N}\exp(\text{sim}(\mathbf{x}_i, \mathbf{z}_j)/\tau)} + \log\frac{\exp(\text{sim}(\mathbf{z}_i, \mathbf{x}_i)/\tau)}{\sum_{j=1}^{N}\exp(\text{sim}(\mathbf{z}_i, \mathbf{x}_j)/\tau)}\right] \quad (1)$$

where $\tau$ is the temperature parameter and the $\text{sim}(\cdot, \cdot)$ is the operator for similarity measurement. Contrastive loss considers that for all $\mathbf{x}_i$, only $\mathbf{z}_i$ is semantically matched (need to pull close), while all $\mathbf{z}_j$ that $j \neq i$ are semantically irrelevant (need to push away), and vice versa. This assumption makes contrastive learning susceptible to data that violates one-to-one correspondence, such as false positives (paired data are weakly-correlated or even no-correlated) and multiple positives (unpaired data may also be semantically consistent).

### 3.2 INPUTS AND ARCHITECTURE

**Inputs.** Considering $N$ videos for pre-training, for audio-visual pair $\{A_i, V_i\}$ in $i$-th video, we sample $N_c$ 2-second paired audio-visual clips. We evenly select $t$ RGB frames for each visual clip and extract log mel spectrograms from audio clips. Finally, the $\{A_i, V_i\}$ is processed to two element paired sequences $a_i = [\mathbf{a}_i^1, \mathbf{a}_i^2, \ldots, \mathbf{a}_i^{N_c}]$ and $v_i = [\mathbf{v}_i^1, \mathbf{v}_i^2, \ldots, \mathbf{v}_i^{N_c}]$, where $\mathbf{a}_i^j \in \mathbb{R}^{h_a \times w_a}$ and $\mathbf{b}_i^j \in \mathbb{R}^{t \times 3 \times h_i \times w_i}$ the audio and visual feature of $j$-th clip in $i$-th video, and $h_a, w_a$ ($h_i, w_i$) represent the length and width of the mel spectrogram (video frame) respectively.

**Architecture.** As illustrated in Figure 1, the visual inputs are encoded by the image encoder $E_i(\cdot)$ followed by the motion encoder $E_m(\cdot)$, while the audio inputs are processed by the audio encoder $E_a(\cdot)$. The feature of audio and frames can be expressed as $\hat{\mathbf{a}}_i^j = E_a(\mathbf{a}_i^j) \in \mathbb{R}^d$ and $\tilde{\mathbf{v}}_i^j = E_i(\mathbf{v}_i^j) \in \mathbb{R}^{t \times d}$, where $E_i(\cdot)$ processes each frame in parallel, and $d$ is the feature dimension. In order to capture motion information in visual data, we add a motion encoder after the image encoder. Specifically, we first add temporal embeddings $\mathbf{e_t} \in \mathbb{R}^{t \times d}$ to the frames feature $\tilde{\mathbf{v}}_i^j$, and

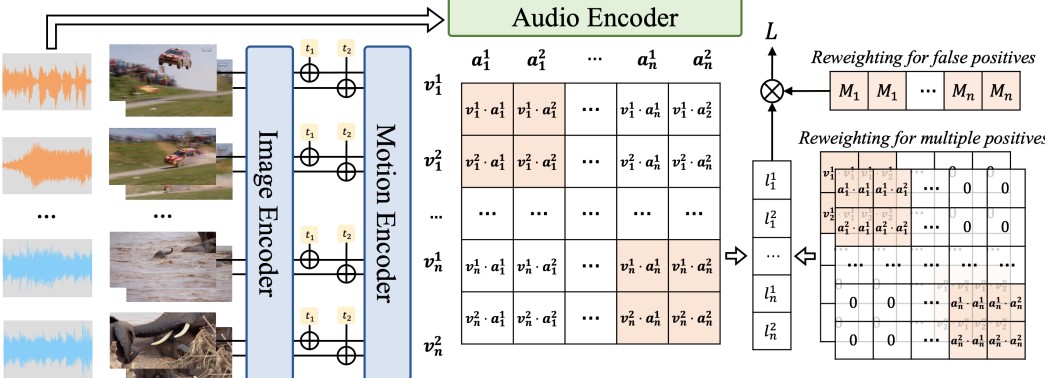

Figure 1: **Illustration of our LiMo.** It comprises an image and motion encoder to extract visual information, alongside an audio encoder for audio representations. Each video input is divided into multiple clips, with the audio and frames of each clip encoded into a shared space. Within this shared space, the audio-visual representations are aligned via a contrastive learning paradigm. To improve the alignment learned from noisy data, we quantitatively measure the likelihood of each sample being a false positive or having multiple positive instances and reweight samples during the final loss calculation.

then input it to our motion encoder to obtain the temporal motion information through interaction between frames. The final visual feature can be expressed as $\hat{\mathbf{v}}_i^j = \mathrm{mean}(E_m(\tilde{\mathbf{v}}_i^j + \mathbf{e_t})) \in \mathbb{R}^d$.

We design the visual encoder that decouples the image and motion encoding process for two reasons: 1) Effectively utilizing the learned knowledge in the pre-trained audio-visual model, which provides alignments between static visual information and audio and largely reduces training costs. 2) Efficiently modeling the spatial and temporal visual information. The transformer (Vaswani et al., 2017) has quadratic computational complexity to the length of input tokens. It is a computationally efficient way of modeling information of each frame in parallel and then capturing the motion information between frames.

### 3.3 ROBUST CLIP-LEVEL AUDIO-VISUAL ALIGNMENT

Previous audio-visual representation methods mainly follow a video-level contrastive loss, which pulls features from the same video close while pushing features of different videos away. To acquire a more precise understanding of the correlation between motion and audio, we further propose clip-level contrastive loss, clips in different videos and the same video are both used for calculating contrastive loss. The main visual difference between videos is object information. Thus, contrastive loss between different videos tends to capture static visual-audio correlation. On the other hand, within a video, the objects of different clips are typically similar. Thus, the motion-audio correlation would be the main clue for distinguishing clips.

To this end, a straightforward method is adopting contrastive loss (Eq. 1) over all $\hat{\mathbf{v}}_i^j$ and $\hat{\mathbf{a}}_i^j$. However, such a learning process is even more susceptible to noisy audio-visual data. On the one hand, the visual and audio information within clips of the same video may be indistinguishable, such as subtle or static motion, repeated audio, and silent clips. These typical situations in videos will lead to multiple positives. On the other hand, false positive video typically means all its clips are audio-visual irrelevant, thus false positives are more common in clip-level contrastive learning.

For robustly learning audio-visual representation from the noisy video clips, we further propose adaptive samples reweighting methods from both the perspectives of multiple positives and false positives:

**Reweighting for Multiple Positives.** Within a video, different clips' visual or audio information may appear undifferentiated. However, the standard contrastive loss function indiscriminately pushes all the unpaired data away, regardless of their distinguishability. To address this, we propose

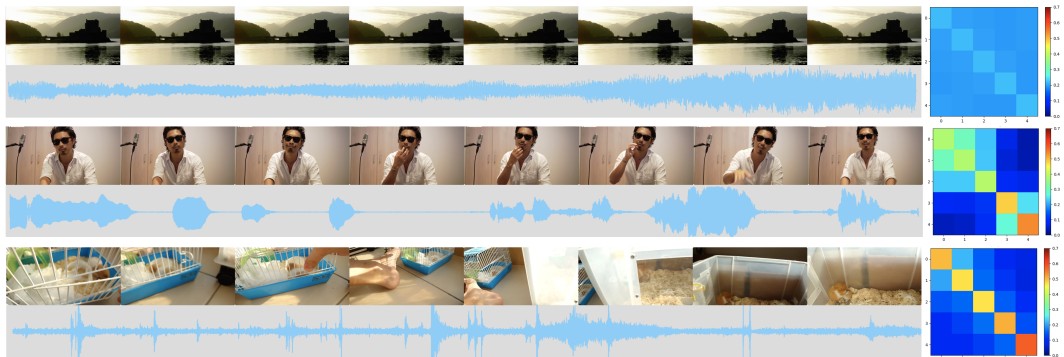

Figure 2: Visualization of the reweighting for multiple positives. We show the videos alongside the distinguishability score $D_i$ between 5 clips evenly sampled from videos.

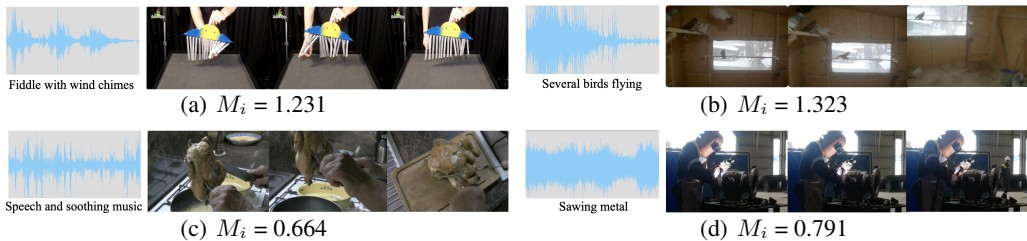

Fiddle with wind chimes

(a) $M_i = 1.231$

Several birds flying

(b) $M_i = 1.323$

Speech and soothing music

(c) $M_i = 0.664$

Sawing metal

(d) $M_i = 0.791$

Figure 3: Visualization of the reweighting for false positives. We display the videos and the corresponding matching confidence $M_i$ between its audio and visual information.

quantitatively measuring the visual and audio distinguishability scores $D_{V_i}, D_{A_i}$ of clips within $i$-th video. Then we combine these two kinds of scores to obtain the overall distinguishability scores matrix $D_i$ for $i$-th video. The calculation process is formulated as follows:

$$D_{A_i}^{j,k} = \hat{\mathbf{a}}_i^j \cdot \hat{\mathbf{a}}_i^k; \quad D_{V_i}^{j,k} = \hat{\mathbf{v}}_i^j \cdot \hat{\mathbf{v}}_i^k \tag{2}$$

$$D_i = \text{Softmax}(\max(\text{s}(D^{A_i}), \text{s}(D^{V_i}))) \tag{3}$$

where $D_{A_i}, D_{V_i}, D_i \in \mathbb{R}^{N_c \times N_c}$ and $D_{A_i}^{j,k}, D_{V_i}^{j,k}$ denote the corresponding distinguishability score between $j$-th and $k$-th clips within $i$-th video, $\text{s}(\cdot)$ normalizes a matrix to $\mathcal{N}(0,1)$ distribution, $\max(\cdot)$ refers to the element-wise maximum operation and $\text{Softmax}(\cdot)$ is the standard softmax function.

As shown in Fig. 2, the weight $D_i$ well reflects the distinguishability between different clips in $i$-th video (considering both visual and audio information). For a video with a still picture, the $D_{V_i}^{j,k}$ are all one, and every value in $D_i$ will be $1/N_c$, which means that all audio clips are positive for all video clips, rather than one-to-one matching. For audios where only the $j$-th clip has sound while other clips are silent or repeated, $D_i^{j,j}$ would be higher than $D_i^{j,k}$ where $k \neq j$. Additionally, the scores $D_i^{t,k}$ between silent clips are identical and higher than $D_i^{j,k}$, effectively highlighting the sounding clip and reflecting the indistinguishability between silent clips.

**Reweighting for False Positives.** For the false positives caused by weakly-correlated or even irrelevant audio-visual pairs, we propose to utilize the audio-visual matching confidence of videos to reweight them. Specifically, we first compute the audio-visual similarity map between videos and use the softmax function to obtain the matching confidence between videos. The lower the matching confidence $M_i$ of $i$-th video, the more likely it is a false positive, and its weight in the learning process should be adaptively reduced. The detailed calculation can be expressed as:

$$M_i = -\frac{1}{2}\left[\frac{\exp((\hat{\mathbf{a}}_i \cdot \hat{\mathbf{v}}_i))}{\sum_{n=1}^N \exp((\hat{\mathbf{a}}_i \cdot \hat{\mathbf{v}}_n))} + \frac{\exp((\hat{\mathbf{v}}_i \cdot \hat{\mathbf{a}}_i))}{\sum_{n=1}^N \exp((\hat{\mathbf{v}}_i \cdot \hat{\mathbf{a}}_n))}\right] \tag{4}$$

where $\hat{\mathbf{a}}_i \in \mathbb{R}^{N_c \times d}$ and $\hat{\mathbf{v}}_i \in \mathbb{R}^{N_c \times d}$ are the concatenation of audio and visual feature of clips in $i$-th video.

**Robust Motion-Audio Alignment Loss.** The above $D_i$ and $M_i$ effectively reflect the likelihood of each sample having multiple positive instances or being false positive. By utilizing these measurements to reweigh samples in the final learning objective calculation, the negative impact of noisy data on contrastive learning can be effectively suppressed.

We first integrate $D_i$ into the contrastive loss calculating process of each clip:

$$L_i^j = -\frac{1}{2}\left[\log\frac{\sum_{k=1}^{N_c} D_i^{j,k}\exp((\mathbf{a}_i^j \cdot \mathbf{v}_i^k)/\tau)}{\sum_{n=1}^{N}\sum_{t=1}^{N_c}\exp((\mathbf{a}_i^j \cdot \mathbf{v}_n^t)/\tau)} + \log\frac{\sum_{k=1}^{N_c} D_i^{j,k}\exp((\mathbf{v}_i^j \cdot \mathbf{a}_i^k)/\tau)}{\sum_{n=1}^{N}\sum_{t=1}^{N_c}\exp((\mathbf{v}_i^j \cdot \mathbf{a}_n^t)/\tau)}\right] \quad (5)$$

For videos whose clips are audio-visual distinguishable, Eq. 5 tends to degenerate into Eq. 1, which emphasizes learning fine-grained motion-audio alignment by distinguishing different clips. For clips that are less distinguishable, Eq. 5 are adaptable to these multiple positive situations, the audio-visual features in these clips would be all pulled together.

After solving the ambiguity problem of multiple positives, we utilize $M_i$ to reweight possible false positives as follows:

$$L = -\frac{1}{NN_c}\sum_{i=1}^{N}\sum_{j=1}^{N_c} M_i \cdot L_i^j \quad (6)$$

The $M_i$ represents the matching confidence between the audio and visual information in $i$-th video. Thus, the final loss of Eq. 6 emphasizes the visual-audio alignment learned from videos with high matching confidence while reducing the importance of videos with lower matching confidence.

# 4 EXPERIMENT

## 4.1 IMPLEMENTATION DETAILS

**Data and Structure.** The clip number $N_c$ of each 10s video is set as 5, and the number of frames $t$ is set as 8. The raw audio waveform of each clip is sampled at 16KHz, and subsequently, a log mel spectrogram with 128 frequency bins is extracted using a 25ms Hamming window with hop length of 10ms. Each RGB frame is resized and center-cropped to 224×224. For the audio and image encoder, we use the same architecture as Girdhar et al. (2023) and initialize our audio and image encoder with its pre-trained weights. The motion encoder is a 2-layer vanilla transformer (Vaswani et al., 2017) encoder with a hidden dimension of 1024 and 4 attention heads. Besides, the motion encoder is zero-initialized to maintain visual object-audio correlation knowledge in the pre-trained weights and stabilize the sample reweighting methods at the beginning of learning. Before the motion encoder, the temporal embeddings $\mathbf{e_t}$ are learnable and zero-initialized.

**Training Configurations.** We utilize the videos from AudioSet-2M (Gemmeke et al., 2017) dataset to pre-train our model. During training, only the motion encoder and the last 4 layers of audio and image encoder are learned to save training costs. The temperature hyper-parameter $\tau$ in Eq. 5 is learnable and initialized as 0.1. We use AdamW (Loshchilov & Hutter, 2017) optimizer with a learning rate initialized $5e^{-5}$ and decayed to $6e^{-5}$ with a cosine schedule. We pre-train our models for 8k iterations, and each batch in training contains 320 videos (1600 clips).

## 4.2 AUDIO-VISUAL CORRELATION

We evaluate the ability to capture audio-visual correlations on audio-visual retrieval over various datasets. Moreover, to specifically verify the correlation between motion and audio, we introduce two motion-specific tasks: audio-based video grounding and lip-speech retrieval. These tasks primarily rely on the visual motion cues in videos to determine the correspondence between audio and visual information.

### 4.2.1 DOWNSTREAM TASKS

**Visual-Audio Retrieval.** We evaluate the visual-to-audio (V2A) and audio-to-visual (A2V) retrieval on AudioSet (Gemmeke et al., 2017), VGGSound (Chen et al., 2020) and MSR-VTT datasets. AudioSet and VGGSound are audio-centric datasets, and their collection focuses on verifying the

Table 1: Audio-visual retrieval results on AudioSet (Gemmeke et al., 2017) and VGGSound (Chen et al., 2020) (Zero-Shot). Frames (times) means the frames and length of video for retrieval. Compared methods includes masked autoencoder-based method (Vanilla AV-MAE (Gong et al., 2022)) and contrastive learning-based methods (CAV, CAV-MAE, CAV-MAE$^{scale+}$ (Gong et al., 2022), ImageBind (Girdhar et al., 2023)). All these models are pre-trained on AudioSet-2M.

| Names | Frames (times) | AudioSet | | | | VGG-Sound | | | |
| | | A2V | | V2A | | A2V | | V2A | |
| | | Top-1 | Top-5 | Top-1 | Top-5 | Top-1 | Top-5 | Top-1 | Top-5 |
|---|---|---|---|---|---|---|---|---|---|
| Vanilla AV-MAE | 10 (10s) | 0.2 | 0.4 | 0.1 | 0.3 | 0.0 | 0.4 | 0.2 | 0.7 |
| CAV | 10 (10s) | 15.5 | 32.7 | 17.4 | 36.1 | 12.4 | 33.2 | 14.2 | 35.2 |
| CAV-MAE | 10 (10s) | 13.5 | 32.5 | 16.1 | 38.6 | 12.1 | 31.6 | 14.7 | 35.3 |
| CAV-MAE$^{scale+}$ | 10 (10s) | 15.1 | 34.0 | 18.8 | 39.5 | 12.8 | 30.4 | 14.8 | 34.2 |
| ImageBind | 8*5 (10s) | 31.8 | 57.2 | 30.1 | 56.1 | 31.9 | 59.8 | 29.6 | 56.4 |
| ImageBind | 8 (2s) | 20.8 | 43.0 | 21.4 | 43.6 | 23.5 | 45.9 | 23.5 | 46.1 |
| LiMo (w/o motion) | 8*5 (10s) | 42.9 | 61.7 | 42.7 | 60.9 | 44.3 | 63.3 | 41.5 | 61.4 |
| LiMo (w/o reweight) | 8*5 (10s) | 46.1 | 68.8 | 46.1 | 68.5 | 45.5 | 71.3 | 44.1 | 69.4 |
| LiMo | 8*5 (10s) | **48.1** | **71.8** | **48.4** | **71.0** | **46.8** | **73.7** | **46.6** | **72.4** |
| LiMo | 8 (2s) | 31.5 | 54.2 | 31.1 | 55.2 | 32.1 | 58.6 | 32.2 | 56.2 |

presence of sounds. MSR-VTT is a visual-centric dataset which is more concerned with visual actions. The retrieval performance on such diverse datasets comprehensively reflects the ability to capture general audio-visual correlations. Following Gong et al. (2022), we use the sampled evaluation subset of 1,725 and 1,525 videos from AudioSet and VGGSound and the whole MSR-VTT evaluation set. We evenly select several 2s clips of each video for each video, and the concatenation of the clips' features is viewed as the representation of the whole video. We encode all videos to the representation space of LiMo and compute the cosine similarity for all audio-visual pairs. The Top-1 and Top-5 metrics are used to measure the retrieval accuracy.

**Motion-specific Audio-Visual Tasks.** We newly propose two motion-specific audio-visual tasks: audio-based video grounding and lip-speech retrieval.

Video grounding (Krishna et al., 2017; Gao et al., 2017; Zhang et al., 2020b) aims to retrieve a video clip from a video to match a language query semantically. This task requires models to fine-grained understand the alignment between language and temporal actions in the video. Similar to video grounding, we introduce an audio-based video grounding task to evaluate the fine-grained motion-audio alignments. This task requires the model to find the visual clip within the 10s video that semantically matches a given 2s audio clip. Following typical video grounding methods (Zhang et al., 2020a;b; Zhao et al., 2021), we employ the mean average Intersection over Union (mIoU) metrics for performance comparison. Refer to the Appendix for the detailed setting of this task.

The lip-speech datasets (Son Chung et al., 2017; Afouras et al., 2018) contain videos of talking faces and the corresponding speech. In these videos, the static visual information is almost the same (all are human faces), and the motion information is the main clue for judging whether the speech is matched. Therefore, the retrieval between visual lip and speech sounds emphasizes the correlation between motions and audio. We employ the retrieval mean average precision (mAP) to evaluate the retrieval performance. Detailed task settings are provided in the Appendix.

### 4.2.2 RESULTS ON AUDIO-VISUAL RETREIEVAL

The audio-visual retrieval results on audio-centric datasets (AudioSet and VGGSound) are reported in Table 1. LiMo showcases dominant performance improvements in audio-visual retrieval compared to other state-of-the-art methods. On the audio-centric datasets, LiMo achieves absolute imptovements of at least 15% top-1 accuracy. Even using fewer frames and shorter video clips, LiMo still outperforms previous methods by a large margin.

Table 2: Audio-visual retrieval results on MSR-VTT. HT-100M denotes HowTo100M (Miech et al., 2019) dataset, and AS-2M denotes AudioSet-2M. The compared baselines contain DAVEnet (Boggust et al., 2019), AVE-Net (Arandjelovic & Zisserman, 2018), AVLnet (Rouditchenko et al., 2020) CAV, CAV-MAE, CAV-MAE$^{scale+}$ (Gong et al., 2022) and ImageBind (Girdhar et al., 2023)

| Methods | Pretrain | A2V | | V2A | |
|---|---|---|---|---|---|
| | | Top-1 | Top-5 | Top-1 | Top-5 |
| DAVEnet | HT100M | 7.6 | 21.1 | 9.3 | 20.7 |
| AVE-Net | HT100M | 12.6 | 26.3 | 11.9 | 25.9 |
| AVLnet | HT100M | 17.8 | 35.5 | 17.2 | 26.6 |
| CAV | AS2M | 6.2 | 17.9 | 10.5 | 25.2 |
| CAV-MAE | AS2M | 7.0 | 18.7 | 10.0 | 26.5 |
| CAV-MAE$^{scale+}$ | AS2M | 7.6 | 19.8 | 13.3 | 29.0 |
| Imagebind | AS2M | 13.9 | 30.8 | 12.5 | 30.7 |
| LiMo (Ours) | AS2M | **20.9** | **41.0** | **20.7** | **40.8** |

Table 3: Audio-based video grounding results on AudioSet (AS) (Gemmeke et al., 2017) and VGGSound (VG) (Chen et al., 2020).

| Methods | AS | VG |
|---|---|---|
| | mIoU | mIoU |
| ImageBind | 17.57 | 18.41 |
| LiMo | **20.51** | **21.16** |

Table 4: Lip-speech retrieval results on LRS2 (Son Chung et al., 2017).

| Methods | V2A | A2V |
|---|---|---|
| | mAP | mAP |
| ImageBind | 6.25 | 7.64 |
| LiMo | **8.54** | **8.74** |

The retrieval performance comparisons on the visual-centric dataset (MSR-VTT) are included in Table 2. LiMo also achieves significantly better performance than all the audio-visual methods even including models trained on 50 times larger datasets. Compared to the advanced ImageBind, LiMo achieves top-1 accuracy relative improvements of 50.3% (13.9% → 20.9%) and 65.6% (12.5% → 20.7%) on audio-to-visual and visual-to-audio retrieval settings.

To specifically learn the effect of our main designs: motion encoder and sample reweighting, we remove each component separately and evaluate their retrieval performance on the audio-centric datasets. The poor performance of LiMo without the motion encoder emphasizes the necessity of capturing motion information in audio-visual learning. The comparison between full LiMo and LiMo without sample reweighting demonstrates that our sample reweighting design can further improve the quality of representation learned from noisy data.

### 4.2.3 RESULTS ON MOTION-SPECIFIC TASKS

In order to more specifically test the alignment quality between motion information and audio in audio-visual representation. We propose two new motion-specific tasks: audio-base video grounding and lip-speech retrieval. As shown in Tab. 3 4, LiMo significantly surpasses the strong baseline ImageBind in these two tasks. These performance improvements indicate that LiMo can better capture and utilize visual motion information to distinguish audio-visual correlations.

Noteworthy, our pre-training data does not contain lip-speech videos. Thus, in the lip-speech retrieval task, models mainly rely on the general motion-audio correlation rather than the semantics alignment of speech. Improvements on such a completely out-of-distribution motion-specific task further demonstrate the deeper understanding and strong generalization of the learned general motion-audio correlations.

### 4.3 AUDIO EVENT RECOGNITION

In addition to enhancing the audio-visual correlated representation, robustly learning motion-audio correlation can also provide more discriminative learning target for audio in the contrastive learning process. To verify the discriminability of audio representations, we freeze the audio encoder and additionally train a linear head for audio event recognition. The linear probing experiment are conducted on three audio event recognition datasets: AudioSet, ESC-50 and Urban-Sound8K. For AudioSet, we use the standard BCE loss

Table 5: Linear evaluation of audio event recognition on AudioSet (AS), ESC-50 and UrbanSound8K (US8K).

| Method | AS | ESC50 | US8K |
|---|---|---|---|
| | mAP | Acc | Acc |
| SM Ensemble | 24.2 | - | - |
| AV-MAE | 24.0 | - | - |
| CAV-MAE | 29.8 | - | - |
| ImageBind | 33.1 | 89.9 | 83.9 |
| LiMo | **33.7** | **91.5** | **85.8** |

Table 6: Ablation study on audio-visual retrieval on AudioSet. The average top-1 metric is reported.

| (a) Sample Reweighting | | (b) Frames | | (c) Video clips | | (d) Motion layers | | (e) Learned layers | |
|---|---|---|---|---|---|---|---|---|---|
| Reweight | Top-1 | $t$ | Top-1 | $N_c$ | Top-1 | n | Top-1 | n | Top-1 |
| w/o $D$&$M$ | 32.91 | 1 | 32.14 | 1 | 32.55 | 0 | 32.59 | 1 | 33.71 |
| w/o $D$ | 33.14 | 4 | 33.46 | 3 | 33.40 | 1 | 33.74 | 2 | 33.74 |
| w/o $M$ | 33.49 | 8 | 33.74 | 5 | 33.74 | 2 | 33.77 | 3 | 34.14 |
| | | | | 7 | 33.77 | 3 | 33.78 | 4 | 34.23 |
| Full | **33.74** | 12 | **33.78** | 9 | **33.91** | 4 | **33.78** | 5 | **34.27** |

function to train the linear head on AudioSet-20K training set, following Gong et al. (2022). For ESC-50 and UrbanSound8K, we report the accuracy under the official n-fold cross-validation.

The experimental results are provided in Tab. 5. LiMo achieves state-of-the-art performance over these audio classification datasets, which further demonstrates the effect of our method in improving the discriminability of audio representations.

## 4.4 Ablation Study

We conduct extensive ablation experiments to study the effect of each component of LiMo. We report the average Top-1 accuracy on the evaluation set of AudioSet. In these experiments, models are trained on the balanced training subset of AudioSet containing only 20K videos (called AudioSet-20K). Besides, by default, the transformer layers of motion encoder are set to 1, and only the last 2 layers of the image and audio encoders are learnable.

**importance of sample reweighting.** We first ablate the distinguishablility scores $D$ and matching confidence $M$ introduced in Sec. 3.3. The results in Tab. 6(a) demonstrate that both measurements contribute to performance, and combining them leads to superior accuracy. Since the training dataset AudioSet-20k in the ablation experiment is relatively small, we also try to remove the sample reweighting method and train LiMo on the full AudioSet-2M, and the performances are reported in Tab. 1. Without sample reweighting, the average top-1 accuracy on AudioSet decreases from 48.3 to 46.1. It proves the importance of sample reweighting when learning correlated audio-visual representations from the noisy large-scale video dataset.

**Effect of frames and video clips.** From the results in Tab. 6(b) and 6(c), we can find that more frame and video clips consistently improve the performance while also significantly increasing the training costs. Furthermore, when only 1 frame is extracted per clip or only 1 clip is sampled per video, the performance will drop dramatically. The former proves the importance of motion information, and the latter shows the necessity of conducting contrastive learning among different clips in the same video.

**Effect of motion layers and learned layers.** We vary the layer number of the motion encoder and the learned layers of the image and audio encoder in Tab. 6(d) and 6(e). Similar to the frame and clips, increasing the number of motion layers or learned layers could improve the retrieval accuracy while bringing higher training costs. Besides, the huge performance gap between without a motion encoder and using a motion encoder (no matter how many layers) in Tab. 6(d) again emphasizes motion information's importance in audio-visual representation learning.

## 4.5 Conclusion

This paper introduces LiMo, a novel framework for audio-visual representation learning. LiMo effectively captures and aligns motion information with audio via robust clip-level contrastive learning. Additionally, to address the challenge of noisy data in web-scale unlabeled videos, we propose a sample reweighting method that adaptively adjusts the weight of each sample based on its probability of being a false positive or containing multiple positive instances. LiMo demonstrates state-of-the-art performance across a range of audio-visual downstream tasks.

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

## A  MORE DETAILS ABOUT MOTION-SPECIFIC TASKS

**Audio-based Video Grounding.**  Video grounding aims to retrieve a video clip from a video to match a language query semantically. This task requires models to fine-grained understand the alignment between language and temporal actions in the video. Similar to this task, we introduce an audio-based video grounding task to evaluate the fine-grained motion-audio alignments. Specifically, for a 10s video, we sample 2s clips every 0.5 seconds. The audio of clips is iterative and used as a query to find the matching visual clip within the 10s video. We calculate the cosine similarity between each audio clip and all the visual clips from the same video, and the visual clip with the highest similarity score is considered as prediction. The used datasets are AudioSet and VGGSound, and the evaluate subset keeps the same to visual-audio retrieval tasks. The mean average Intersection over Union (mIoU) metrics are used for performance comparison.

**Lip-Speech retrieval.**  The lip-speech dataset typically contains videos of talking faces and the corresponding speech. Since the static visual information in these videos is almost the same (all are human faces), the motion information is the main clue for judging whether the speech is matched. We select 1000 3s lip-speech videos from lip-speech datasets LRS2. Similar to the general audio-visual retrieval task, the human speech (audio) and lip video (visual) are encoded into LiMo's audio-visual representation space, and the mAP retrieval metric is used to evaluate the retrieval accuracy Noteworthy, our pre-training data does not contian such lip-speech video data, thus the lip-speech retrieval mainly relies on the general motion-audio correlation, rather than the semantics of speech. This out-of-distribution motion-specific task further reflects understanding and generalization of general motion-audio correlations.

## B  LIMITATIONS AND FUTURE WORK

Our method provides a plain yet effective network structure for modeling motion information, and for saving training costs and maintain the acquired static visual-audio correlation knowledge, we only tune the last few layers of the audio encoder and image encoder. More sophisticated network structure design and how to capture motion information more effectively while maintaining existing static visual-audio correlation knowledge will be interesting directions. Moreover, as discussed in Sec. 4.2.3 more motion-specific audio-visual tasks are also very important for more in-depth verification of audio-visual representation capabilities.

