# OpenReview forum: "Listen to Motion: Robustly Learning Correlated Audio-Visual Representations"
_ICLR.cc/2024/Conference — ICLR 2024 Conference Withdrawn Submission_

### Official Review · Reviewer_2yxk · 2023-10-23

**Soundness:** 2 fair
**Presentation:** 3 good
**Contribution:** 2 fair
**Rating:** 5
**Confidence:** 4

**Summary:**

This paper deals with the problem of learning audio-visual representations from unlabeled consumer videos and proposes a method for weighing segment-level features based on two types of "distinguishability" scores. (Note: I use a "clip" for the whole video, say 10 seconds in the experiment, and a "segment" for a part of the "clip," say 2 seconds, for clarification, which is different from the paper.) The first distinguishability is based on intra-clip, inter-segment, and intra-modal similarity scores, and the second one is based on clip-wise audio-visual similarity scores. Based on those distinguishability scores, the proposed method softly defines positive and negative segment pairs for contrastive learning. The proposed method also introduces an additional layer for aggregating features along with the temporal domain. Experimental evaluations with AudioSet, VGG-Sound, and LRS2 demonstrate that the proposed method outperformed several popular baselines in the tasks of audio-visual retrieval, visual grounding, and lip-speed retrieval.

**Strengths:**

1. The problem dealt with in this paper is significant and relatively unexplored. Text-related multi-modal / cross-modal representation learning is largely explored, and we can find several strong baselines, such as CLIP and Chirp. However, audio-visual representation learning has no solid foundation models, indicating room for improvement in this research topic.

2. The proposed method is simple and easy to implement, preferable for practitioners interested in this method.

3. The experimental evaluations demonstrate that the proposed method outperformed several standard baselines in this research topic.

4. The current paper is well-written and easy to follow.

**Weaknesses:**

1. I am unsure whether the proposed method has sufficient technical contributions.
  - The "motion encoder" is just a temporal aggregation of short-time features, which does not contribute to improving technical novelty.
  - SImilarity-based assignments of positives and negatives have already been discussed in multi-modal / cross-modal representation learning, e.g., CrossCLR [Zolfaghari+ ICCV2021] and adaptive contrastive learning [Nguyen+ EMNLP2022].
  - I understand that the proposed method differs from previous methods in terms of detailed implementations. However, the underlying ideas seem to be the same. The technical novelty of the proposed method against them and empirical evaluations would be required.

2. The proposed method heavily relies on audio-visual representation pre-learning. Without this pre-training, Equations (2) and (4) will produce meaningless similarity scores. From this viewpoint, I am unsure whether the experimental result presented in Table 1 is fair.

3. I am unsure whether the evaluation in Table 1 is fair from another viewpoint. The proposed method has a large margin from the other previous method even if one of the additional components (feature aggregation and reweighting) is missing, which implies that there might be a secret. It might be the existence of pre-training, the frame setting, or tricks for training.

**Questions:**

All the issues presented in the Weakness part are the questions to the authors. I look forward to providing an illuminating response to each case.

**Details Of Ethics Concerns:**

None.

---

### Official Review · Reviewer_bjXv · 2023-10-31

**Soundness:** 2 fair
**Presentation:** 2 fair
**Contribution:** 3 good
**Rating:** 5
**Confidence:** 3

**Summary:**

The paper presents a method for audio-visual representation learning using a robust clip-level contrastive learning. This type of contrastive learning uses a sample re-weighting method where the weights are the likelihood of a sample being a false positive or containing multiple false positives. A video is divided into Nc number of paired audio-visual clips. The audio clip is passed through an audio encoder whereas  the video clip is passed through a image encoder, temporal encoding is added which is then passed to the motion encoder. For the case of multiple positives in a video, audio features of two clips are combined together to get an audio score and similarly video features are combined together to get a video score. The scores are normalized and then the max of the score is taken. Finally, softmax is applied over the maximum score. This score is helpful in understanding the distinguishability between the different clips in the same video. For the case of false positives, an audio-visual similarity map is taken and softmax function is used to obtain the matching confidence between videos. The two scores are then finally used in the contrastive loss function.

**Strengths:**

1. The method has a significant improvement in the performance on the task of audio-visual retrieval.
2. The introduction is well-written and clearly explains the difference between visual object and motion information and why the motion-audio alignment can be important for learning.

**Weaknesses:**

1. While the discussion of using temporal embedding and motion encoding process is appreciated in Section 3.2, it does not seem to be a very strong module in itself from observing Table 6d. Increasing the number of motion layer does not show a significant change in the performance. At the same time, it is a bit unclear if the split between image and temporal encoder is in any way related or helpful to the paper's main idea of robust contrastive learning. To prove if this module is necessary, an ablation experiment is needed which shows a comparison between the performance of a video-based model and image-temporal encoding process.

2. The ablation study is not very strong. Table 6b and Table 6c do not give much information. Table 6b shows that increasing the number of frames shows a performance improvement, however, it seems to be obvious as the model is trying to learn from a video. Increasing the number of frames would give more temporal information and give a stronger signal to the model to learn the temporal motion of a video. At the same time, Table c shows that increasing the number of video clips also help which also seems obvious as it can be interpreted as a means to increase the training data/augmentation.

3. It is a bit unclear as to how the softmax over the audio-visual similarity map can accurately find the false positives as it is still computing the audio visual similarity of the same audio visual pair. The same audio-visual similarity is also used in the contrastive learning loss.

4. For calculating the score of multiple positives, can there be a scenario where the $D_{A_{i}}^{j,k}$ of two sounding clips and $D_{A_{i}}^{j,k}$ two silent clips is the same? Can there be a scenario where the audio-visual clips are distinguishable but the scores are same?

**Questions:**

1. Some clarity is needed as to what false positive means in the context of the work. In the paper, they are defined as irrelevant audio-visual pairs, however, there can be some explanation as to what does an irrelevant audio-visual pair mean. Do they mean pairs with noisy audio-visual correspondence or no correspondence?

2. Some examples can be given as to what false positives and multiple positives mean in the paper.

3. In what scenarios, can the score of multiple positives will be same even if the audio-visual pair is distinguishable?

---

### Official Review · Reviewer_5gkS · 2023-11-04

**Soundness:** 1 poor
**Presentation:** 2 fair
**Contribution:** 2 fair
**Rating:** 1
**Confidence:** 1

**Summary:**

The paper introduces Listen to Motion (LiMo), a new approach that incorporates motion information to refine the correlation between audio and dynamic visual elements. LiMo works by extracting temporal visual semantics to foster interaction between video frames while preserving the static visual-audio correlations of prior models. It improves audio-visual alignment by differentiating between different video clips and introducing a method to identify and reweight false positive or multiple positive instances. Extensive experiments are conducted on several retrieval and motion-specific tasks.

**Strengths:**

+ The proposed adaptive reweighted contrastive loss addresses the noisy data issue in audio-visual self-supervised learning.

+ The experiments show that the proposed method achieves superior performance over compared approaches on several tasks, including Audio-visual retrieval, audio-base video grounding, and Lip-speech retrieval.

**Weaknesses:**

The contributions of the work are overclaimed, and some very important relevant works are missing. A lot of claims and statements are clearly wrong.

1. In the paper, the authors claimed that

(a) "Previous methods mainly model the static “object” information from a few frames while lacking the ability to capture and align the important temporal “motion” information";

(b) "Although these methods achieve promising performance on different audio-visual downstream tasks, they either lack modeling of visual motion information or explicit learning of motion audio alignment, which significantly constrains their upper bound in capturing audio-visual correlations";

(c) "Previous audio-visual representation methods mainly follow a video-level contrastive loss, which pulls features from the same video close while pushing features of different videos away. To acquire a more precise understanding of the correlation between motion and audio, we further propose cliplevel contrastive loss, clips in different videos and the same video are both used for calculating contrastive loss."

However, self-supervised audio-visual learning using temporal synchronization beyond semantic correspondence matching is not new. In 2018, Andrew and Alexei [1] proposed a method to learn audio and visual representations by training a neural network to predict whether video frames and audio are temporally aligned. They paired temporally shifted audio with visual frames from the same video to build negative samples. Concurrently, Korbar et al. [2] adopted contrastive learning for audio-visual self-supervised learning, using both easy and hard negative samples. "Easy negatives" are pairs where the video frames and audio come from two different videos. "Hard negatives" are pairs taken from the same video, but with at least a half-second time gap between the audio sample and the visual clip. These two pioneering works are well-known in the field, and the first work is probably the most cited self-supervised audio-visual learning work. I was surprised that the authors ignored these works and made these invalid statements.

[1] Owens, Andrew, and Alexei A. Efros. "Audio-visual scene analysis with self-supervised multisensory features." Proceedings of the European conference on computer vision (ECCV). 2018.

[2] Korbar, Bruno, Du Tran, and Lorenzo Torresani. "Cooperative learning of audio and video models from self-supervised synchronization." Advances in Neural Information Processing Systems 31 (2018).

2. About the audio-visual noisy data issue, the authors claimed that

(a) "Despite their impressive performance, two key issues still limit the further development of audio-visual representations: ... 2) The unlabeled web-scale video data are noisy. In a video, the visual information is limited in the camera perspective, while the audio can originate from all directions. Consequently, not all visual objects make sounds, and not all sound sources are visible in the video.
This unavoidable noisy data compromises the quality of the learned representations."

(b) "the web-scale video data for training is very noisy. However, previous audio-visual representation methods lack the analysis and design to alleviate the adverse effect of the noisy data."

Morgado et al. [3] have already explored the issue of false positive noisy data. They proposed a weighted contrastive learning loss to down-weigh the contribution of false positives to the overall loss. The main idea of the proposed method is very similar to what they did.
The authors have also ignored this work, which clearly contradicts their claim of novelty.

[3] Morgado, Pedro, Ishan Misra, and Nuno Vasconcelos. "Robust audio-visual instance discrimination." Proceedings of the IEEE/CVF Conference on Computer Vision and Pattern Recognition. 2021.

Considering these facts, I do not think the work can be accepted.

I wonder why the authors did not evaluate their method on video action recognition (see [3]) and audio-visual classification (see CAV-MAE paper), which are standard evaluation tasks for audio-visual learning.

**Questions:**

Please see the Weaknesses.